# Carvedilol Selectively Stimulates βArrestin2-Dependent SERCA2a Activity in Cardiomyocytes to Augment Contractility

**DOI:** 10.3390/ijms231911315

**Published:** 2022-09-26

**Authors:** Jennifer Maning, Victoria L. Desimine, Celina M. Pollard, Jennifer Ghandour, Anastasios Lymperopoulos

**Affiliations:** Laboratory for the Study of Neurohormonal Control of the Circulation, Department of Pharmaceutical Sciences, Nova Southeastern University College of Pharmacy, Fort Lauderdale, FL 33328, USA

**Keywords:** β-adrenergic receptor, βarrestin2, cardiomyocyte, contractility, G protein-coupled receptor, SERACA2a, signal transduction, SUMOylation

## Abstract

Heart failure (HF) carries the highest mortality in the western world and β-blockers [β-adrenergic receptor (AR) antagonists] are part of the cornerstone pharmacotherapy for post-myocardial infarction (MI) chronic HF. Cardiac β_1_AR-activated βarrestin2, a G protein-coupled receptor (GPCR) adapter protein, promotes Sarco(endo)plasmic reticulum Ca^2+^-ATPase (SERCA)2a SUMO (small ubiquitin-like modifier)-ylation and activity, thereby directly increasing cardiac contractility. Given that certain β-blockers, such as carvedilol and metoprolol, can activate βarrestins and/or SERCA2a in the heart, we investigated the effects of these two agents on cardiac βarrestin2-dependent SERCA2a SUMOylation and activity. We found that carvedilol, but not metoprolol, acutely induces βarrestin2 interaction with SERCA2a in H9c2 cardiomyocytes and in neonatal rat ventricular myocytes (NRVMs), resulting in enhanced SERCA2a SUMOylation. However, this translates into enhanced SERCA2a activity only in the presence of the β_2_AR-selective inverse agonist ICI 118,551 (ICI), indicating an opposing effect of carvedilol-occupied β_2_AR subtype on carvedilol-occupied β_1_AR-stimulated, βarrestin2-dependent SERCA2a activation. In addition, the amplitude of fractional shortening of NRVMs, transfected to overexpress βarrestin2, is acutely enhanced by carvedilol, again in the presence of ICI only. In contrast, metoprolol was without effect on NRVMs’ shortening amplitude irrespective of ICI co-treatment. Importantly, the pro-contractile effect of carvedilol was also observed in human induced pluripotent stem cell (hIPSC)-derived cardiac myocytes (CMs) overexpressing βarrestin2, and, in fact, it was present even without concomitant ICI treatment of human CMs. Metoprolol with or without concomitant ICI did not affect contractility of human CMs, either. In conclusion, carvedilol, but not metoprolol, stimulates βarrestin2-mediated SERCA2a SUMOylation and activity through the β_1_AR in cardiac myocytes, translating into direct positive inotropy. However, this unique βarrestin2-dependent pro-contractile effect of carvedilol may be opposed or masked by carvedilol-bound β_2_AR subtype signaling.

## 1. Introduction

Sarco(endo)plasmic reticulum Ca^2+^-ATPase (SERCA)-2a is a crucial, for contractile function, calcium-handling protein expressed in the mammalian myocardium and its downregulation is one of the molecular hallmarks of chronic heart failure (HF) [1]. SERCA2a activation is part of the signaling mechanism by which the β_1_-adrenergic receptors (ARs) increase cardiac contractility [2]. Agonist-bound β_1_ARs however, like most G protein-coupled receptors (GPCRs), undergo functional desensitization/internalization due to the actions of βarrestin1 or -2, two universal GPCR adapter proteins, following the phosphorylation of agonist-occupied receptors by GPCR-kinases (GRKs) [3]. During receptor endocytosis, the two βarrestins initiate their own wave of G protein-independent signaling [4]. Among the cellular processes βarrestins regulate is protein SUMO (small ubiquitin-like modifier)-ylation, which generally increases protein stability/levels in tissues, including the heart [5]. We have reported that cardiac β_1_AR-activated βarrestin2, but not βarrestin1, promotes SERCA2a SUMOylation and activity, thereby directly increasing cardiac contractility [6,7]. βarrestin2 also exerts anti-inflammatory and anti-apoptotic effects in the post-myocardial infarction (MI) heart [8].

Among the three currently US FDA-approved β-blocker drugs for human chronic HF: bisoprolol, metoprolol, and carvedilol [9,10], the latter is the only one reported to be a βarrestin-“biased” agonist [11,12] and to upregulate SERCA2a in cardiac myocytes via gene transcriptional and anti-oxidative mechanisms [13,14,15]. Metoprolol (β_1_AR-selective) lacks this effect [15]. Carvedilol is also known to activate Ca^2+^ signaling in CNS neurons in a βarrestin2-dependent manner [16]. Carvedilol is pharmacologically a βAR non-subtype-selective inverse agonist, also blocking α_1_ARs [17]. However, the adult human heart expresses β_1_ARs:β_2_ARs:α_1_ARs at a ratio of 70:20:10 (in terms of total AR percentages), i.e., the β_1_AR subtype is by far the most predominant cardiac AR subtype expressed [2,17]. On the other hand, carvedilol is ~7-fold more potent at human β_1_AR than at β_2_AR (also ~2-fold more potent at β_1_AR than at α_1_AR) [17]. Taken together, these findings strongly suggest that carvedilol is relatively β_1_AR-selective, at least in the human heart (both non-failing and failing). In the present study, we posited that carvedilol may selectively stimulate cardiac β_1_AR-dependent βarrestin2 signaling to SERCA2a SUMOylation and activation, which would translate into enhanced contractile function.

## 2. Results

### 2.1. Carvedilol, but Not Metoprolol, Induces βarrestin2 Interaction with Cardiac SERCA2a, Enhancing SERCA2a SUMOylation

Carvedilol is known to be a βarrestin-“biased” agonist [11,12] and several of its well-documented beneficial effects in chronic HF have been attributed to activation of βarrestin-dependent signaling [11,12,18]. Interestingly, carvedilol, but not metoprolol, has been reported to enhance cardiac SERCA2a levels and activity [13,14,15] and appears to exert minimal (if any) negative inotropy when given chronically for human HF treatment [19]. Thus, to examine whether βarrestin2-dependent SERCA2a SUMOylation/activity plays any role in these unique, among the β-blocker drugs, effects of carvedilol, we overexpressed βarrestin2 in H9c2 cardiomyocytes (Figure 1A) and compared carvedilol with metoprolol head-to-head in their effects on SERCA2a. Neither drug had any effect on SERCA2a SUMOylation in the absence of βarrestin2 (i.e., in control AdGFP-transfected H9c2 cells, which express only βarrestin1 endogenously) (Figure 1A) [7,20]. In βarrestin2-expressing cardiomyocytes however, carvedilol acutely stimulated βarrestin2 interaction with SERCA2a, leading to increased SUMOylation of the latter (Figure 1B,C). In contrast, metoprolol was incapable of inducing βarrestin2 interaction with SERCA2a or of increasing SERCA2a SUMOylation (Figure 1B,C). This suggests that the mechanism by which carvedilol uniquely stimulates cardiac SERCA2a levels/activity might be the induction of β_1_AR-dependent βarrestin2 recruitment to SERCA2a and enhanced subsequent SERCA2a SUMOylation.

### 2.2. Carvedilol, but Not Metoprolol, Increases βarrestin2-Dependent Cardiac SERCA2a Activity, but β_2_AR Inhibits This Effect

Next, we examined whether the βarrestin2-dependent SERCA2 SUMOylation selectively induced by carvedilol indeed results in enhanced SERCA2a activity in H9c2 cardiomyocytes, which would be expected based on the results of Figure 1. To our surprise however, carvedilol treatment did not enhance SERCA activity neither in control (AdGFP), nor in βarrestin2-overexpressing cells (Figure 2). The same was true for metoprolol (Figure 2). Given that carvedilol, unlike metoprolol, binds both β_1_- and β_2_AR subtypes, we speculated that the fact it increases SUMOylation without increasing activity of SERCA2a might be due to some functional competition between carvedilol-bound β_1_AR and carvedilol-occupied β_2_AR signaling to SERCA2a. Indeed, that proved to be the case, since pretreatment with the β_2_AR-specific inverse agonist ICI 118,551 (ICI) allowed carvedilol to acutely increase SERCA2a activity in βarrestin2-overexpressing cardiomyocytes (Figure 2). Metoprolol was again without effect. Thus, it seems that signaling by the β_2_AR subtype somehow prevents βarrestin2-mediated SERCA2a SUMOylation from increasing the pump’s activity in cardiomyocytes.

### 2.3. Carvedilol, but Not Metoprolol, Stimulates Cardiomyocyte βarrestin2-Dependent Contractility in a β_2_AR-Sensitive Manner

We next sought to test the impact of the increased βarrestin2-dependent SERCA2a activity by carvedilol on cardiomyocyte contractility. Because H9c2 cardiomyoblasts do not contract, we used neonatal rat ventricular myocytes (NRVMs) instead, overexpressing βarrestin2 via adenoviral-mediated transfection. First, we confirmed that, similarly to the H9c2 cardiomyocytes above, neither metoprolol, nor carvedilol affected SERCA2a SUMOylation in control NRVMs that express almost exclusively βarrestin1 endogenously (Figure 3A), but carvedilol acutely stimulated βarrestin2 interaction with SERCA2a, leading to increased SUMOylation of the latter in the βarrestin2-expressing myocytes (Figure 3B,C). Metoprolol again did not stimulate the SERCA2a-βarrestin2 interaction and did not affect SERCA2a SUMOylation (Figure 3B,C). We also observed that carvedilol potentiated SERCA activity only in the presence of ICI 118,551 in βarrestin2-expressing NRVMs (Figure 3D), similarly to the H9c2 cells above (Figure 2), whereas metoprolol did not affect SERCA activity irrespective of concomitant β_2_AR antagonism (Figure 3D). Thus, after essentially recapitulating all our findings from H9c2 cells, we tested the effect of these β-blockers on NRVM contractility. Carvedilol alone was incapable of increasing contractility of NRVMs despite βarrestin2 overexpression. However, upon pretreatment with ICI, it resulted in significant enhancement of contractility compared to basal conditions (0.1% DMSO, no stimulation) (Figure 3E). Metoprolol did not affect contractility regardless of concurrent β_2_AR blockade (Figure 3E). This finding strongly suggests that carvedilol, but not metoprolol, can increase cardiomyocyte contractility via βarrestin2-dependent SERCA2a SUMOylation and stimulation, as long as it is prevented from binding to the β_2_AR, i.e., this effect of carvedilol is strictly β_1_AR-dependent.

### 2.4. Carvedilol, but Not Metoprolol, Stimulates βarrestin2-Dependent Contractility in Human Fully Differentiated Cardiac Myocytes

To verify that the findings above hold true in bona fide human cardiac myocytes, we took advantage of the iCell^®^ cardiomyocyte preparation available from Fujifilm Cellular Dynamics. This is essentially human cardiac (ventricular) myocytes (CMs) differentiated from human induced pluripotent stem cells (hIPSCs). After culturing the cells for two weeks, we transfected them with adenovirus encoding for βarrestin2 (encoding also for GFP as a marker of gene transduction in a bi-cistronic manner [7]), so as to overexpress βarrestin2 (Figure 4A). 48 h post-adenoviral infection, we applied the drugs to measure the contractile responses of the hIPSC-CMs. As shown in Figure 4B, metoprolol was without effect on cell shortening, with or without concomitant β_2_AR blockade with ICI. In contrast, carvedilol alone did produce a weak but nevertheless significant increase in the fractional shortening of βarrestin2-overexpressing hIPSC-CMs, a response that was further augmented by ICI pretreatment (β_2_AR blockade) (Figure 4B). As expected, the β_1_AR full agonist dobutamine elicited a far more robust (~3 times larger than carvedilol alone) contractile response (Figure 4B). Taken together, these results indicate that carvedilol can increase contractility of human fully differentiated cardiac myocytes via β_1_AR-activated βarrestin2, and, unlike rat neonatal cardiomyocytes, concomitant β_2_AR blockade is not necessary for this pro-contractile effect of carvedilol to be manifested.

## 3. Discussion

In the present study, we have uncovered a unique pharmacological action of the β-blocker carvedilol: unlike metoprolol and other β-blocker drugs commonly used in cardiovascular clinical practice, ridden with negative inotropy, carvedilol not only does not reduce cardiomyocyte contractility but actually increases it via the β_1_AR-βarrestin2-SERCA2a signaling axis (Figure 5). This may translate into a direct positive inotropic effect of this β-blocker drug per se in the heart, which can further enhance (via baroreflex activation) its anti-hypertensive, vasodilatation-mediated effect, as well. Of course, this awaits confirmation in animal models in vivo and, ultimately, in patients.

SERCA2a has a prominent role in the myocardium regulating (enhancing) both relaxation (by removing cytosolic Ca^2^) and contraction (by increasing the availability of Ca^2+^ for release into the cytosol). Thus, its activation is among the most important signaling mechanisms of catecholamine-induced positive inotropy via the cardiac β_1_AR [1,21]. Therefore, it comes as no surprise that cardiac SERCA2a downregulation, both in terms of protein levels and pump activity, is a molecular hallmark of human HF [1]. Although initially thought to signal and function similarly to the β_1_AR, cardiac β_2_AR signaling and function are now known to be substantially different [22]. For instance, β_2_AR is anti-apoptotic and very minimally pro-contractile (i.e., exactly the opposite of β_1_AR) and it can also couple to G_i/o_ (inhibitory/other G) proteins [2,22,23]. β_2_AR also undergoes different alterations in human HF vs. the β_1_AR [24]. Specifically, in human HF, β_1_AR is desensitized and selectively downregulated (i.e., total cellular receptor levels reduced due to reduced receptor protein synthesis), whereas the β_2_AR is also desensitized and dysfunctional, but its plasma membrane density is unaltered [10,24].

βarrestin1 is far more abundant than βarrestin2 in the myocardium of various species (human, murine, rat) [6,7,25,26]. Considered for a long time as functionally interchangeable, the two βarrestins are now known to exhibit remarkable functional divergence, both in heterologous cell lines in vitro and in various tissues in vivo, including in the heart [18]. More specifically, βarrestin1 exerts negative inotropy by decoupling the cardiac β_1_AR from G_s_ protein/cyclic 3′, 5′-adenosine monophosphate (cAMP) (i.e., classic functional desensitization of the cardiac β_1_AR) [2,7,27,28]. In contrast, βarrestin2 knockout mice display significantly worse overall mortality and cardiac apoptosis, more inflammation/macrophage infiltration of the heart, and, importantly, significantly worse ejection fraction/cardiac contractility post-MI, compared to HF wild type mice [8]. Additionally, the human polymorphic Arg389 β_1_AR variant, known to be hyperfunctional and more pro-contractile in the heart than its Gly389 counterpart, interacts with βarrestin2 more efficiently and to a larger extent than the Gly389 β_1_AR variant [29,30]. Importantly, βarrestin2 directly enhances cardiac contractility by (a) not affecting (desensitizing) β_1_AR signaling via the G_s_–cAMP pro-contractile pathway, and (b) directly binding SERCA2a and promoting its SUMOylation, a process known to acutely enhance SERCA2a protein stability and calcium-pumping activity (Figure 5) [1,6,7]. This acute effect of cardiac βarrestin2 might complement long-term stimulatory effects of this adapter protein on cardiac SERCA2a, since βarrestin2-“biased” agonism has been reported to transcriptionally upregulate cardiac SERCA2a in a mouse model of dilated cardiomyopathy [31]. Thus, βarrestin2 attenuates adverse remodeling (apoptosis and inflammation) while directly enhancing cardiac contractility (via SERCA2a potentiation) (Figure 5) [7,28].

Among the three FDA-approved β-blocker agents for chronic HF treatment, carvedilol has been postulated to be the most superior/preferred agent [9], since it appears to exert favorable effects on cardiac contractility [19]. Restoration of adrenal GRK2-α_2_AR-catecholamine secretion axis and suppression of norepinephrine release from cardiac sympathetic nerve terminals contribute to the beneficial effects of β-blockers in HF [32,33]. Carvedilol has several unique, among the β-blocker class, properties: significant vasodilation, as α_1_AR-blocker, which minimizes (via the baroreceptor reflex mechanism) the negative effect of cardiac βAR blockade on cardiac output and may also inhibit the cardiotoxic maladaptive effects of cardiac α_1_ARs (maladaptive hypertrophy) [2,17,34,35,36]; potent anti-oxidant, anti-endothelin, and anti-proliferative agent [9,37,38]; and, unlike all other β-blockers (including metoprolol) that upregulate β_1_AR in the failing human heart, carvedilol does not reverse β_1_AR downregulation in the failing heart despite exerting complete βAR blockade [17,35,39]. This is probably due to the fact that carvedilol has also unique sympatholytic properties, keeping norepinephrine levels in HF patients at bay [35]. This sympatholysis, coupled with its potent inverse agonism at cardiac βARs (especially at the β_1_AR), and with its significant affinity for all three types of cardiac ARs (β_1_AR, β_2_AR, α_1_AR), culminates in a powerful, almost complete adrenergic blockade by carvedilol in human HF [17,35].

Given that the negative inotropy exerted by β-blockers limits exercise capacity and reduces quality of life of HF patients, the β-blocker agent(s) resulting in the smallest reduction in cardiac contractility are preferred for HF with reduced ejection fraction (HFrEF) treatment. Carvedilol has been suggested to lack a direct negative effect on cardiac contractility and may directly increase it in human HF [19], possibly because it increases SERCA2a levels and activity [13,14,15]. Additionally, carvedilol stimulates intracellular Ca^2+^ signaling via βarrestin2 (but not βarrestin1) in native central nervous system neurons, an effect that might also be mediated by SERCA upregulation [16]. Our present study focused on comparing the pro-contractile effects of carvedilol vs. metoprolol, since only these two agents, out of the three currently approved for chronic HF (i.e., not bisoprolol), are known to activate βarrestin signaling [11,12]. Our findings strongly suggest that carvedilol, but not metoprolol, enhances SERCA2a SUMOylation and activity via βarrestin2 in cardiac myocytes, resulting in direct augmentation of cardiac contractility (Figure 5). Metoprolol lacks this effect, perhaps because it stimulates βarrestin1 (rather than βarrestin2) binding to the cardiac β_1_AR and βarrestin1 cannot stimulate SERCA2a SUMOylation or activity [7]. Thus, carvedilol, either alone or combined with cardiac-specific βarrestin2 gene therapy, can provide relief for acute HF decompensation episodes during the long-term treatment of chronic human HF. Importantly, given that βarrestin2 inhibits apoptosis and inflammation in the post-MI heart (Figure 5) [7,8], carvedilol’s putative positive inotropy should be safe for the myocardium. Our findings on carvedilol’s pro-contractile effects in cardiac myocytes are consistent with a recent study reporting that this β-blocker uniquely enhances skeletal muscle contractility, as well [40]. Although this effect of carvedilol was attributed to βarrestin1, rather than βarrestin2, and to β_2_AR signaling, instead of β_1_AR, the specific underlying signaling mechanisms were not investigated [40]. In addition, this study was done exclusively in transgenic mice in vivo, in which skeletal muscle β_2_AR activation by endogenous catecholamines could very well have skewed the observed effects of carvedilol on contractility.

There are some obvious caveats associated with our present findings. First, carvedilol requires βarrestin2 to stimulate contractility, which is normally minimally expressed in the adult heart [6,25,26]. Moreover, carvedilol causes robust sympatholysis and does not reverse cardiac β_1_AR downregulation in human HF. Taken together, these observations suggest that the weak positive inotropic effect of carvedilol manifested in cardiomyocytes in vitro may be masked or significantly blunted/offset in human HF in vivo. Indeed, carvedilol was found ineffective at stimulating βarrestin-dependent contractility in isolated primary murine cardiomyocytes [26], probably because these myocytes, like their human counterparts, exhibit minimal βarrestin2 protein expression (contrary to the very robust βarrestin1 expression) [7,26]. Nevertheless, βarrestin2 levels may change (increase) in failing human hearts, in an effort to compensate for the functional decline; thus, carvedilol’s positive inotropy via βarrestin2 may become manifested in HF patients. Indeed, a recent study hinted at βarrestin2 upregulation in failing rat hearts [41]. Another pitfall emerging from our present work is the antagonism the cardiac β_2_AR subtype seems to exert on carvedilol’s effects on SERCA2a. The underlying mechanism(s) for this is presently unknown but one possible explanation would be that perhaps the carvedilol-occupied β_2_AR subtype favors βarrestin1 over βarrestin2 activation in the heart. βarrestin1 not only does not promote SERCA activity, like βarrestin2 does, but also, indirectly, decreases it by desensitizing βAR signaling towards cAMP-protein kinase A (PKA)-dependent SERCA2a activation [7]. Another potential explanation for the opposite effects of β_2_AR (vs. β_1_AR) on carvedilol-induced SERCA2a SUMOylation and activation is that the β_2_AR, unlike the β_1_AR, can also couple to Gi/o proteins [23]. Gi/o protein signaling dampens cAMP levels in the heart, thereby reducing SERCA2a activity and contractility, which would work against carvedilol’s actions. Nevertheless, in the adult human cardiomyocyte, wherein the β_1_AR subtype is far more abundant than the β_2_AR subtype [17], this negative effect of the β_2_AR on carvedilol’s pro-contractile effects via SERCA2a might be negligible. Our data in hIPSC-CMs (Figure 4), in which carvedilol was capable of increasing contractility without concomitant β_2_AR blockade, strongly hint at this possibility. Of note, H9c2 cardiomyoblasts and neonatal cardiomyocytes express much higher β_2_AR densities than adult cardiomyocytes [7,42,43], which could explain why β_2_AR blockade is necessary to unmask carvedilol’s effects on SERCA2a and contractility in H9c2 cells and in NRVMs but not necessary at all in hIPSC-CMs. Finally, carvedilol’s pro-contractile effect is quite weak, especially when compared to the robust cardiomyocyte response to standard cardiac β_1_AR agonists like dobutamine (Figure 4). This is totally expected however, since carvedilol, unlike dobutamine, cannot stimulate the Gs protein/cAMP signaling pathway, which robustly increases cardioymyocyte contractility [2,21].

Our present study has two major limitations: (a) Although we used physiologically relevant cells (cardiomyoblasts, neonatal cardiomyocytes, and stem cell-derived differentiated cardiomyocytes), these cells were not, nonetheless, primary bona fide adult cardiomyocytes and were studied exclusively in vitro; and (b) Our findings obviously require confirmation in in vivo models and settings. Nevertheless, given that the cell models we employed were differentiated cardiac myocytes, including, importantly, human cardiomyocytes, our present findings are quite likely to hold true in adult human primary cardiomyocytes in vivo, as well.

In conclusion, carvedilol, uniquely among β-blockers, increases cardiomyocyte contractility via β_1_AR-activated, βarrestin2-dependent SERCA2a SUMOylation and potentiation. Thus, contrary to the conventional wisdom that all β-blockers are negative inotropes, which complicates their use and limits their dosages in human chronic HF therapy, carvedilol may have direct positive inotropic actions in the human failing myocardium. This would give carvedilol the special advantage of being suitable for use in chronic human HFrEF complicated by frequent acute decompensated HF episodes.

## 4. Materials and Methods

### 4.1. Materials

All drugs and chemicals were from Sigma-Aldrich (St. Louis, MO, USA).

### 4.2. Cell Culture and Transfections

The H9c2 rat cardiomyoblast cell line was purchased from American Type Culture Collection (Manassas, VA, USA) and cultured as previously described [7]. NRVMs were purchased from Lonza (Cat. #R-CM-561; Lonza Group Ltd., Basel, Switzerland) and plated and cultured according to the supplier’s manual. hISPC-CMs (iCell^®^ cardiomyocytes) were purchased from Fujifilm Cellular Dynamics, Inc. (Cat. #R1105; FCDI, Madison, WI, USA) and plated and cultured in the supplier’s provided culture medium according to their instructions. Recombinant adenoviruses encoding for green fluorescent protein (AdGFP) and for rat wild type full-length βarrestin2 (Adβarr2, also encoding for GFP via a bi-cistronic construct) were constructed as described previously [7,44]. Briefly, transgenes were cloned into shuttle vector pAdTrack-CMV, which harbors a CMV-driven GFP, to form the viral constructs by using standard cloning protocols. The resultant adenoviruses were purified using two sequential rounds of CsCl density gradient ultracentrifugation, as described previously [44]. To verify the identity of hIPSC-CMs as cardiomyocytes, immunofluorescence with an antibody against human NKX2.5 (cardiomyocyte marker) (Cat. #SAB2501264; Sigma Aldrich) was performed as described previously [44]. To confirm overexpression of βarrestin2 in hIPSC-CMs, GFP fluorescence (green) was visualized under a fluorescence microscope.

### 4.3. Immunoprecipitation/Western Blotting for SERCA SUMOylation Determination

Protein concentration was determined and equal amounts of protein per sample were used for immunoprecipitation (IP) or Western blotting. SERCA2a was immunoprecipitated by overnight incubation of extracts with an anti-SERCA2 antibody (Cat. #4388; Cell Signaling Technology, Danvers, MA, USA), attached to Protein A/G-Sepharose beads (Sigma-Aldrich). The IPs were then subjected to immunoblotting for βarrestins with an anti-βarrestin1/2 antibody (Cat. #sc-28869; Santa Cruz Biotechnology, Santa Cruz, CA, USA), for SUMO-1 (Cat. #4930; Cell Signaling Technology), and for SERCA2, to confirm IP of equal amounts of endogenous SERCA2a among the various treatments. Immunoblots were revealed by enhanced chemiluminescence (ECL, Life Technologies, Grand Island, NY, USA) and visualized in the FluorChem E Digital Darkroom (Protein Simple, San Jose, CA, USA), as described previously [7]. Densitometry was performed with the AlphaView software (Protein Simple) in the linear range of signal detection (on non-saturated bands).

### 4.4. SERCA Activity

Maximal SERCA activity was measured as described [7,15,45]. Briefly, lysates were prepared and assayed in 10 mM Tris, pH 7.5, 100 mM KCl, 5 mM MgCl_2_, 5 mM Na_2_ATP, 0.1 mM CaCl_2_, 0.2 mM NADH, 1.5 mM trisodium phosphoenolpyruvate, 15 units/mL pyruvate kinase, and 36 units/mL lactate dehydrogenase. Total ATPase activity was assayed by monitoring the rate of loss of A340 after addition of the membrane preparation to a thermostatically controlled (37 °C) cuvette in a spectrophotometer. Background ATPase activity was determined in the absence of ATP. Ca^2+^-independent ATPase activity was assayed in the presence of 10 mM EGTA, instead of Ca^2+^, and subtracted from the total ATPase activity to derive the Ca^2+^-dependent ATPase (SERCA) activity. SERCA activity was calculated as nmol of inorganic phosphate (P_i_) produced per min per mg of protein.

### 4.5. Cell Shortening Measurements

NRVMs or hIPSC-CMs were plated on laminin-coated glass coverslips, in a buffer with serial Ca^2+^ concentration titrations. The myocytes were then placed in MEM (Sigma-Aldrich) containing 1.2 mM Ca^2+^, 2.5% (*v/v*) FBS, and 1% penicillin/streptomycin. The pH was adjusted to 7.0 in 5% CO_2_ by the addition of NaHCO_3_. After 1 h at 5% CO_2_ and 37 °C, medium was replaced with serum-free MEM containing 0.1 mg/mL bovine serum albumin (BSA). Myocytes attached to coverslips were bathed in 700 μL of air/37.

C-equilibrated, 20 mM HEPES (pH 7.4)-buffered Medium 199 (Sigma-Aldrich), containing 1.8 mM calcium and 100 nM N106 [7] to acutely stimulate SERCA2a SUMOylation, and used within 2 h of plating. Contractility (cell shortening) was measured at a pacing frequency of 2 Hz immediately upon drug (or DMSO) addition into the medium via a digital videocamera connected to the microscope, as described [46,47]. Briefly, to calculate the drug-induced % shortening, the baseline (without drug) length change was subtracted from the drug-induced one for each drug/agent, and then plotted as % change (increase in cell length shortening) over no drug stimulation. For example: an average Δlength of ~5 μm was observed both with and without Meto, indicating 0% contractility increase, but Dob addition resulted in average Δlength of ~7.5 μm from a baseline average Δlength of ~5 μm (without Dob), i.e., ~50% contractility increase on average. The measurements were done for 6–8 different myocytes per drug tested and the whole series of drugs was tested three independent times (i.e., after three independent myocyte platings). The operator was blinded as to the identity of the drug/agent added each time.

### 4.6. Statistical Analysis

Data are generally expressed as mean ± SEM. Unpaired 2-tailed Student’s *t* test and one- or two-way ANOVA with Bonferroni test were generally performed for statistical comparisons, unless otherwise indicated. For most 3-group statistical comparisons, Dunnett’s test using SAS version 9 software (SAS Institute Inc., Cary, NC, USA) was used, as well. For all tests, a *p* value of <0.05 was generally considered to be significant.

## Figures and Tables

**Figure 1 ijms-23-11315-f001:**
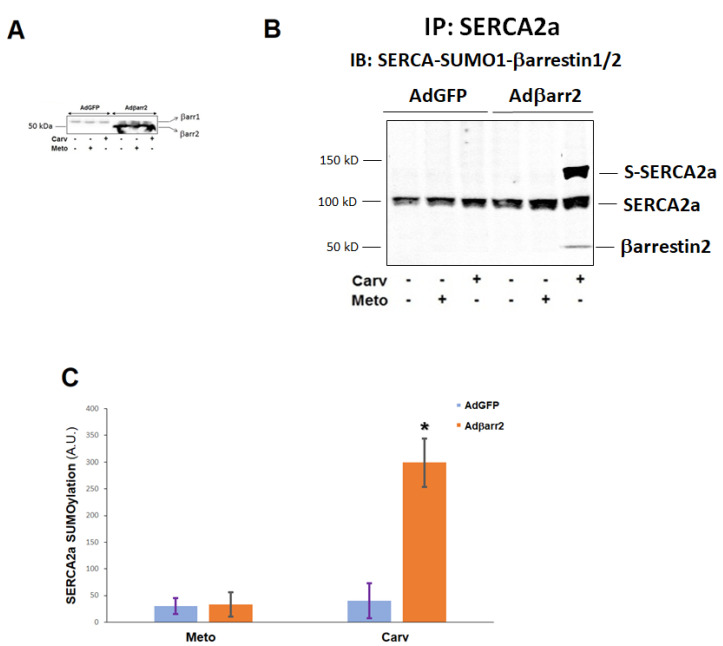
β-blockers and βarrestin2-dependent SERCA2 SUMOylation in H9c2 cardiomyocytes. (**A**) Immunoblotting for βarrestins in control (infected with adenovirus encoding for green fluorescent protein, AdGFP) or in H9c2 cardiomyocytes infected with adenovirus encoding for βarrestin2 (Adβarr2) to confirm βarrestin2 overexpression in the latter cells. H9c2 cardiomyocytes express only βarrestin1 (βarr1) endogenously at detectable levels (see AdGFP lanes and Refs. [7,20]). Shown is a representative blot of three independent experiments. (**B**,**C**) Immunoblotting for βarrestins, SUMO-1 and SERCA2a in SERCA2a immunoprecipitates (IPs) prepared from these cells and treated with vehicle (0.1% DMSO), 1 μM carvedilol (Carv), or 1 μM metoprolol (Meto) for 20 min. Representative blots are shown in (**B**) and SERCA2a SUMOylation quantitation, as measured by densitometry, is shown in (**C**). IB: Immunoblotting; IP: Immunoprecipitation; S-SERCA2a: SUMOylated SERCA2a; A.U.: Arbitrary (densitometric) units. Only βarrestin2 (not βarrestin1) could be detected in the SERCA2a IPs and only in Adβarr2-infected cells treated with carvedilol. *****, *p* < 0.05, vs. any other treatment/cell clone; n = 3 independent experiments.

**Figure 2 ijms-23-11315-f002:**
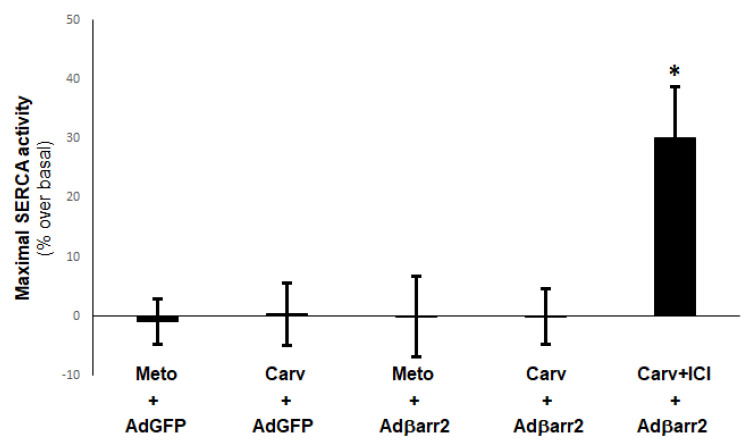
β-blockers and SERCA2a activity in βarrestin2-overexpessing H9c2 cardiomyocytes. Maximal SERCA activity in microsomes isolated from βarrestin2-expressing (Adβarr2) or control (AdGFP) H9c2 cardiomyocytes and treated with 1 μM metoprolol (Meto), 1 μM carvedilol (Carv), or 1 μM carvedilol in the presence of 10 μM ICI 118,551 (Carv + ICI) for 30 min. *****, *p* < 0.05, vs. any other group; n = 3 independent measurements/transfection group/treatment.

**Figure 3 ijms-23-11315-f003:**
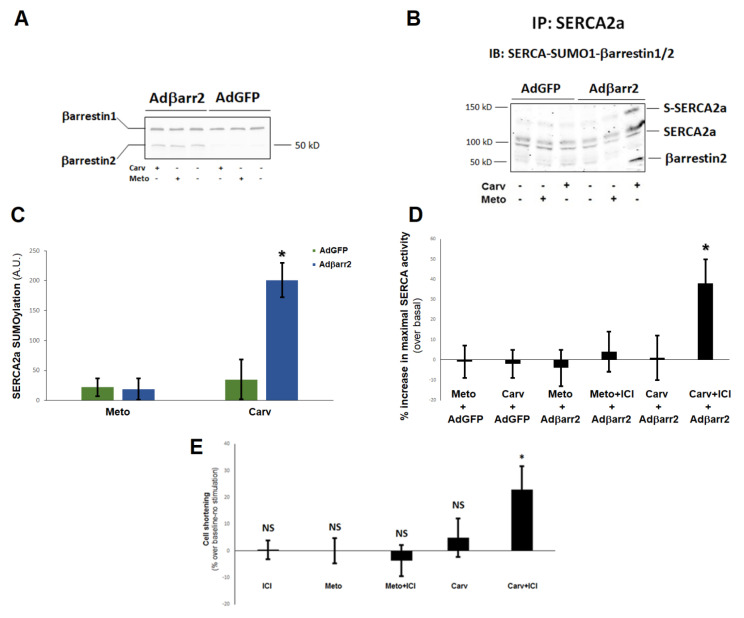
β-blockers and βarrestin2-dependent SERCA2a function and contractility in NRVMs. (**A**) Immunoblotting for βarrestins in control (AdGFP) or in NRVMs expressing βarrestin2 (Adβarr2) to confirm βarrestin2 transgene induction. Like H9c2 cardiomyocytes, NRVMs also appear to express only βarrestin1 endogenously at detectable levels (see AdGFP lanes). Shown is a representative blot from three independent experiments. (**B**,**C**) Immunoblotting for βarrestins, SUMO1 and SERCA2a in SERCA2a IPs prepared from these cells and treated with vehicle (0.1% DMSO), 1 μM carvedilol (Carv), or 1 μM metoprolol (Meto) for 20 min. Representative blots are shown in (**B**) and SERCA2a SUMOylation quantitation, as measured by densitometry, is shown in (**C**). S-SERCA2a: SUMOylated SERCA2a; A.U.: Arbitrary (densitometric) units. As in H9c2 cardiomyocytes, only βarrestin2 could be detected in SERCA2a IPs and only in Adβarr2-infected cells treated with carvedilol. *****, *p* < 0.05, vs. any other treatment/cell clone; n = 3 independent experiments. (**D**) Maximal SERCA activity potentiation in microsomes isolated from βarrestin2-expressing (Adβarr2) or control (AdGFP) NRVMs. ICI: 10 μM ICI 118,551; Meto: 1 μM metoprolol; Carv: 1 μM carvedilol, for 30 min. *****, *p* < 0.05, vs. any other group; n = 3 independent measurements/transfection group/treatment. (**E**) Cell shortening response of βarrestin2-overexpressing NRVMs treated as in (**D**). *****, *p* < 0.05, vs. any other treatment; NS: Not significant vs. basal (0.1% DMSO) at *p* = 0.05; n = 3 independent measurements (in triplicate)/treatment group.

**Figure 4 ijms-23-11315-f004:**
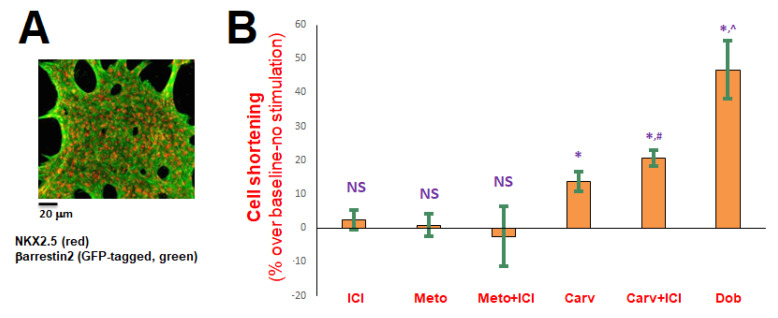
Cell shortening response of βarrestin2-overexpressing hIPSC-CMs. (**A**) Co-immunofluorescence performed on culture day 16 (48 h post-adenoviral infection) for NKX2.5 (cardiac myocyte marker, red) and for GFP (green) to confirm βarrestin2 overexpression (the adenovirus encoding for βarrestin2 also encoded for GFP, see Section 4.2 for details). (**B**) ICI: 10 μM ICI 118,551; Meto: 1 μM metoprolol; Carv: 1 μM carvedilol; Dob: 1 μM dobutamine. *****, *p* < 0.05, vs. baseline (0.1% DMSO); ^#^, *p* < 0.05, vs. Carv alone; ^, *p* < 0.05, vs. any other treatment; NS: Not significant vs. basal (0.1% DMSO) at *p* = 0.05; n = 3 independent measurements in 6–8 different myocytes per drug tested.

**Figure 5 ijms-23-11315-f005:**
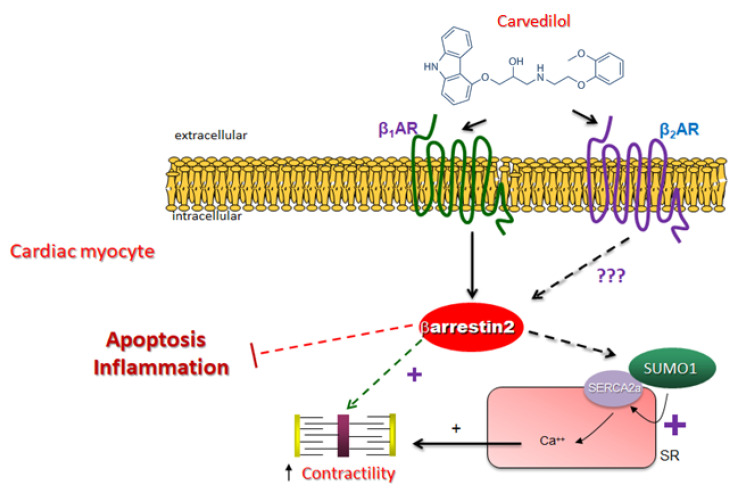
Schematic illustration of carvedilol’s positive inotropic effect via stimulation of the β_1_AR-βarrestin2-SERCA2a SUMOylation axis in cardiac myocytes. See text for details. SR: Sarcoplasmic reticulum; SUMO1: Small ubiquitin-like modifier-1; ???: Effect/signaling pathway unknown.

## Data Availability

All source data files are available upon request to the correspondence author.

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
