# Peer review of "Carvedilol Selectively Stimulates βArrestin2-Dependent SERCA2a Activity in Cardiomyocytes to Augment Contractility"

_ijms, 2022, doi:10.3390/ijms231911315_

Round 1

Reviewer 1 Report (Previous Reviewer 2)

Please check if the only band shown in the last original figure are really from this gel image.

Author Response

Please check if the only band shown in the last original figure are really from this gel image.

Author response: Although this comment is quite vague (exactly which last original figure and which gel image is the reviewer referring to here?), we have double-checked our original blots and can confirm they correspond to the blots shown in the figures of our manuscript. 

Reviewer 2 Report (New Reviewer)

The manuscript is nicely written with smooth logic of the study. With additional control experiments on ICI alone and measurement on Ca2+ transient/SR Ca2+ load, the manuscript would be acceptable.

Major issues:

·         Fig 2, Fig 3D, Fig 3E: have the authors tested Carv+AdGFP+ICI to make sure that beta-arr2 is really involved in the SERCA2 activity changes? Or just ICI alone, or ICI+Ad-beta-arr2? Could ICI alone or ICI+beta-arr2 cause changes of SERCA2 activity? Then carvedilol will not have significant effects on SERCA2 activity.

·         Fig 4B: ICI should be tested alone as a control.

·         Has Ca2+ transient/SR Ca2+ load been tested? These are more direct proof for SERCA2a contribution on contractility.

·         The method to measure contractility/cell shortening is questionable. Supplementary video files may be needed to show changes with drugs. Also details of how to calculate the contractility to obtain Fig 3E/Fig 4B are needed.

Minor:

·         Line 21, is it “this b2AR subtype on carvedilol-occupied……”?

·         Line 171: please clarify that these hiPSC-CMs are ventricular like or atrial- or nodal-like cells.

Round 2

Reviewer 1 Report (Previous Reviewer 2)

It is my question if the S-SERCA2a band in Figure 3B is original or a copy of another blot.

This is not the original image, but a processed one. You should replace this one with a real one.

Author Response

We apologize but we are still not entirely sure what the reviewer means here. The blot in question (Fig. 3C top panel) is cropped from an original blot image (the full uncropped original version is now provided in the Supplementary file) and definitely not a copy of another blot. Perhaps some confusion arose from the fact that, upon blotting for SERCA2a (i.e., with an anti-SERCA2a antibody), both non-SUMOylated and SUMOylated (running at a slightly higher molecular weight) versions of the protein are detected (Fig. 3C middle panel), but obviously, when blotting for S-SERCA2a (with an anti-SUMO1 antibody to detect SERCA2a-attached SUMO-1), only the SUMOylated protein (S-SERCA2a) is detected (Fig. 3C top panel). We hope this now is clear to this reviewer and our apologies again for any confusion inadvertently caused.

Reviewer 2 Report (New Reviewer)

My concerns are all answered properly. 

One last comment is that the experiment/result of ICI alone as control done in previous work was not cited when the results were mentioned. This causes the problem for the first round review. Also methods used in a manuscript should have enough details for other people to repeat the experiments. If there is word limit, the detailed methods should be in Supplementary. Citing previous works is fine but not enough.

Author Response

We thank this reviewer for his/her invaluable comments that have helped us improve the quality of our original manuscript significantly. In response to his/her latest comments, we have elaborated on our methodology in section 4.5 (lines 400-408, highlighted in yellow) of our newly revised manuscript. This should be (and we hope it is) satisfactory now.

Round 3

Reviewer 1 Report (Previous Reviewer 2)

There is also a problem with Figure 3A, in addition to Figure 3C. In Figure 3A, it is apparently a fake figure. There are three lanes of βarrestin1 and βarrestin2 (in red circle), while there are four lanes of Adβarr2 treatment (in blue circle) in the same gel. Apparently, the three lanes of βarrestin1 and βarrestin2 do not match the four lanes of Adβarr2 treatment.

Round 4

Reviewer 1 Report (Previous Reviewer 2)

No more comments.

This manuscript is a resubmission of an earlier submission. The following is a list of the peer review reports and author responses from that submission.

Round 1

Reviewer 1 Report

The publication Carvedilol Selectively Stimulates βArrestin2-Dependent SERCA2a Activity in Cardiomyocytes to Augment Contractility  is very high quality and well written. It provides new original results, but in my opinion these results should also be verified in vivo.

-           

-         -  In line 16 is carvedilol state as β-blocker compare to line 52 where it is state as βAR non-subtype selective inverse agonist. Acording what you say: „In the present study, we posited that carvedilol may selectively stimulate cardiac β1AR-dependent βarrestin2 signaling to SERCA2a SUMOylation and activation,  which would translate into enhanced contractile function“, I suppose that carvedilol is agonist of βAR.

-          - Graph representation of immunoblotting is missing.

-          - Blots of normalizing protein are missing.

-         -  I would recommend adding, if not possible, other proteins in the next publication, e.g. beta-adrenergic receptors, apoptotic pathways, PKA and others.In figure legends it would be propriate to state exact statistical method which was used.

-          - Quantification of alterations in protein levels of all mentioned βAR as well as αAR  would be very enriching for this publication.

Author Response

The publication Carvedilol Selectively Stimulates βArrestin2-Dependent SERCA2a Activity in cardiomyocytes to Augment Contractility  is very high quality and well written. It provides new original results, but in my opinion these results should also be verified in vivo.

  1. In line 16 is carvedilol state as β-blocker compare to line 52 where it is state as βAR non-subtype selective inverse agonist. Acording what you say: „In the present study, we posited that carvedilol may selectively stimulate cardiac β1AR-dependent βarrestin2 signaling to SERCA2a SUMOylation and activation,  which would translate into enhanced contractile function“, I suppose that carvedilol is agonist of βAR.

Author response: We thank this reviewer for the overall kind and positive comments about the quality of our work. This is a pertinent remark raised by this reviewer. Carvedilol is pharmacologically categorized as a “b-blocker” (although it also blocks a1-adrenoceptors) and is an inverse agonist with respect to G protein activation by bARs, i.e., it suppresses constitutive G protein activation by bARs (see Ref. 17 of our manuscript). However, with respect to bAR-elicited barrestin signaling, it is considered an agonist, a so-called barrestin-“biased” agonist, exactly because it activates barrestin signaling without activating G protein signaling (e.g., see Refs. 11 & 40 of our manuscript for pertinent reviews on this topic). We apologize for the confusing nature of the nomenclature but we hope this is clear(er) now to this reviewer.

  1. Graph representation of immunoblottingis missing.

Author response: We are not sure which immunoblots exactly the reviewer refers to here. The SERCA SUMOylation quantitation is presented in a bar graph (Fig. 1C), so we guess he/she refers to the barrestin2 interaction with SERCA2a. We chose not to quantify this in a graph because we could only detect this interaction upon carvedilol treatment of Adbarr2 cells. It was absent in all other treatment conditions, i.e., it was an all-or-none occurrence, so quantitative comparisons among the treatment groups were unnecessary.  

  1. Blots of normalizing protein are missing.

Author response: Again, we are not sure which immunoblots the reviewer is referring to. For the SERCA2a IPs to measure SUMOylation and barrestin2 interaction (Fig. 1B), the immunoprecipitated SERCA2a itself serves as the normalizing protein, which is shown (Fig. 1B). If the reviewer is referring to the blot of Fig. 1A: since barrestin2 is not detectable endogenously (see AdGFP lanes), there was no point in showing normalizing protein for exogenous barrestin2 (over)expression in our view. Besides, endogenous barrestin1 levels can serve as the normalizing protein, shown as the upper (faint) band in the blot of Fig. 1A, which were indeed equal among all treatments. We hope this satisfies now this reviewer.

4, I would recommend adding, if not possible, other proteins in the next publication, e.g. beta-adrenergic receptors, apoptotic pathways, PKA and others.In figure legends it would be propriate to state exact statistical method which was used.

Author response: We thank this reviewer for this excellent suggestion, which we will follow in future studies. All statistical methods used are described in detail in Section 4.6 of “Materials and Methods”. 

  1. Quantification of alterations in protein levels of all mentioned βAR as well as αAR  would be very enriching for this publication.

Author response: This is an interesting comment by this reviewer but we have no evidence or data that bAR or aAR protein levels are changed neither by the exogenous barrestin2 overexpression, nor by any of our drug treatments. Besides, the drug treatments were acute (short-lived, only 20 min for SERCA SUMOylation, 30 min for SERCA activity, and a few seconds for cell shortening determinations), i.e., hardly enough time for detectable changes in the levels of any cellular protein to occur.

Reviewer 2 Report

The topic of this work is interesting. Although the authors investigated different cardiomyocytes, all studies were carried out on normal cells and not heart failure-related pathophysiological cells. There is no obvious mechanism that translates carvedilol's stimulation of SERCA2a SUMOylation, SUMOylation, and activity through the β1AR in cardiac myocytes to pro-contractile effects of carvedilol in heart failure. The present manuscript also contains some problems.

1.    Bisoprolol, metoprolol, and carvedilol are β-blocker drugs approved by the FDA for chronic HF. In this study, why were only metoprolol and carvedilol used?

2.    Can metoprolol or carvedilol affect the expression of arrestin2?

3.    Line 28: this statement is incorrect.

4.    A treatment of 1 uM metoprolol in the presence of 10 uM ICI 118,551 should be added to Figure 2.

5.    Part 2.1 and part 2.2 examined the mechanism of β-blocking drugs and the cell function in two different cardiomyocytes. The molecular mechanisms should also be studied in neonatal rat ventricular myocytes (NRVMs). Considering that H9c2 cardiomyoblasts do not contract, it is unclear why this cell line is still used and why they did not use NRVMs to begin with.

6.   The effects of Meta and ICI on neonatal rat ventricular myocyte contractility were synergistic, but no significant differences were observed in Figure 3. There may be significant differences between ICI only, Meto only, and Meto+ICI. Could the author explain the effect of Meto+ICI on neonatal rat ventricular myocyte contractility?

7.    Figure 4A should include Co-immunofluorescence staining in the Ad-GFP control treatment.

8.    In the Discussion, the authors should delve deeper into their findings.

9.    The paper needs extensive corrections to grammar and punctuation errors.

Author Response

The topic of this work is interesting. Although the authors investigated different cardiomyocytes, all studies were carried out on normal cells and not heart failure-related pathophysiological cells. There is no obvious mechanism that translates carvedilol's stimulation of SERCA2a SUMOylation, SUMOylation, and activity through the β1AR in cardiac myocytes to pro-contractile effects of carvedilol in heart failure. The present manuscript also contains some problems.

  1. Bisoprolol, metoprolol, and carvedilol are β-blocker drugs approved by the FDA for chronic HF. In this study, why were only metoprolol and carvedilol used?

Author response: We thank this reviewer for the overall kind and positive comments about the quality of our work. This is an excellent point, and we thank this reviewer for raising it. We tested only carvedilol and metoprolol (not bisoprolol), simply because only these two b-blockers, out of the three approved for chronic HF, have been shown to activate barrestin signaling (e.g., see Refs. 11 & 12 of our manuscript).  In other words, bisoprolol, to our knowledge at least, is not a barrestin-“biased” agonist. We apologize for not having clarified this in the original version of our manuscript, but we clearly state it now in the revised manuscript (lines 247-250, highlighted in yellow). We hope this is clear now to this reviewer and we thank him/her again for this invaluable remark.

  1. Can metoprolol or carvedilol affect the expression of arrestin2?

Author response: This is another excellent remark by this reviewer. We do not have any data or evidence that either drug affects barrestin2 (or barrestin1, for that matter) expression, nor are we aware of any study in the literature claiming such an effect. In any case, even if metoprolol or carvedilol can affect barrestin expression, this possibility must be excluded in our present study, given that our drug treatments were acute and short-lived (<30 min of drug exposure only), i.e., hardly enough time for barrestin expression levels to change.

  1. Line 28: this statement is incorrect.

Author response: We have rephrased that sentence accordingly in the revised abstract (see lines 28-29 of the revised manuscript, highlighted in yellow).

  1. A treatment of 1 uM metoprolol in the presence of 10 uM ICI 118,551 should be added to Figure 2.

Author response: We thank the reviewer for this interesting suggestion but, given that metoprolol did not induce barrestin2 interaction with SERCA2a or affect SERCA2a SUMOylation (Fig. 1), ICI co-treatment probably would have not affected metoprolol-induced SERCA activity, either. The data on the effect of Meto+ICI on contractility (Figs. 3 & 4) corroborate this, as well. Nevertheless, we did not add ICI to metoprolol in our experiments of Fig. 2, because we deemed this co-treatment unnecessary. Our apologies for this but we hope the reviewer understands and agrees with our rationale behind this choice.

  1. Part 2.1 and part 2.2 examined the mechanism of β-blocking drugs and the cell function in two different cardiomyocytes. The molecular mechanisms should also be studied in neonatal rat ventricular myocytes (NRVMs). Considering that H9c2 cardiomyoblasts do not contract, it is unclear why this cell line is still used and why they did not use NRVMs to begin with.

Author response: The reason behind using H9c2 cells for our mechanistic/signaling studies was purely logistic: H9c2 is an immortalized cell line, easily maintained & propagated in culture, and readily transfectable. Additionally, these cells express the entire molecular machinery of a bone fide cardiomyocyte cell, i.e., they are an ideal in vitro cardiomyocyte cell model for mechanistic/signaling studies, which is exactly what they are used for by almost every lab worldwide. In contrast, NRVMs and, even more so, hIPSC-CMs are notoriously hard to isolate and culture, as they are extremely sensitive to their cell culture/maintenance environment, and, most importantly, they do not proliferate (primary cells). Thus, with NRVMs and human CMs, it is virtually impossible to procure/culture enough viable/healthy cells at the numbers necessary to perform the mechanistic/signaling experiments of Figs. 1 & 2 with proper statistical power. That`s why we performed our mechanistic studies in H9c2 cells and our contractility studies in NRVMs and human CMs. We hope the reviewer understands now.

  1. The effects of Meta and ICI on neonatal rat ventricular myocyte contractility were synergistic, but no significant differences were observed in Figure 3. There may be significant differences between ICI only, Meto only, and Meto+ICI. Could the author explain the effect of Meto+ICI on neonatal rat ventricular myocyte contractility?

Author response: Our apologies, but we are not sure we understand what the reviewer means here. We did not observe any statistically significant differences among ICI, Meto, and Meto+ICI treatments on cell shortening of either NRVMs (Fig. 3) or hIPSC-CMs (Fig. 4), i.e., their effects were indistinguishable from that of 0.1% DMSO (vehicle), meaning that metoprolol-occupied β1AR does not increase contractility, regardless of barrestin2 expression or β2AR activity statuses. In other words, metoprolol acts as a pure, neutral βAR antagonist (without intrinsic agonistic/pro-contractile activity) in cardiac myocytes with respect to contractility.

  1. Figure 4A should include Co-immunofluorescence staining in the Ad-GFP control treatment.

Author response: Thank you for this suggestion but, respectfully, we do not think this is necessary, given that the co-IF staining was done to demonstrate the feasibility and success of the Adbarr2 infection of the hIPSC-CMs, i.e., to illustrate the exogenous expression of barrestin2 (tagged with GFP). We hope the reviewer understands.  

  1. In the Discussion, the authors should delve deeper into their findings.

Author response: Done, thank you. We have elaborated a bit more in the “Discussion” of our revised manuscript (e.g., see added sentence on metoprolol: lines 253-255, highlighted in yellow).

  1. The paper needs extensive corrections to grammar and punctuation errors.

Author response: Done, thank you. We have proofread the entire text of our manuscript and corrected all errors we were able to find. 

Round 2

Reviewer 2 Report

In this study, NRVMs were purchased from Lonza (Cat. #R-CM-561; Lonza Group Ltd, Basel, Switzerland). They are not hard to isolate and culture. There are many studies to  perform the mechanistic/signaling experiments using CMs isolated enzymatically from 1-3 day old Sprague-Dawley rat ventricles.